# KnItLM: Weaving Knowledge into Instruction-Tuned Language Models via Continual Pretraining and Merging

## Abstract

RAG has become the de facto method for incorporating new, corpus-specific knowledge into an instruction following LLM (Instruct LLM). Although RAG-based prompting improves factual grounding, it fails when retrieval is incorrect or incomplete, leading to hallucinations. Finetuning methods such as RAFT Zhang et al. (2024b) and PA-RAG Bhushan et al. (2025) enhance RAG by ingesting new knowledge into the parameters of the model, but require generating massive amount of synthetic QA that covers the entire corpus. Continued Pre-Training (CPT) on the text corpus avoids the need for comprehensive synthetic data generation but breaks the instruction following capabilities of an Instruct LLM, necessitating instruction fine-tuning (IFT) post CPT. However, IFT is costly and may be infeasible due to the unavailability of an instruction tuning corpus. In this work, we propose KnItLM-KNowledge IngesTion via LoRA Merging. Instead of doing CPT on the Instruct LLM, KnItLM performs CPT with Low-Rank Adapters (LoRA) on its corresponding base LLM to infuse new knowledge. These knowledge-infused LoRA weights are then merged with the Instruct LLM, imparting new knowledge without impacting their instruction following capabilities. KnItLM avoids expensive instruction fine-tuning and relies on model merging (Ilharco et al. (2023)) to infuse the new knowledge into the Instruct LLM without destroying its instruction following capabilities. Empirical results show that KnItLM significantly improves the performance of RAG by taking accuracy from 54.17% to 79.26% for retrieval failure cases. In addition, the proposed method achieves superior performance to existing approaches while requiring substantially less training data.

## 1 Introduction

'*Base LLMs*' trained on an enormous amount of textual data have immense knowledge, but lack instruction-following capabilities. This necessitates post-training, which typically involves massive Instruction Fine-Tuning (IFT) (Ouyang et al., 2022; Shengyu et al., 2023) followed by RLHF (Schulman et al., 2017; Rafailov et al., 2023; Pandey et al., 2024). '*Instruct LLMs*' (model obtained after post–training) have achieved remarkable success across general-purpose tasks (Brown et al., 2020; Wei et al., 2022). However, their deployment in specialized domains such as question answering over technical or confidential policy documents, shifts the emphasis from general reasoning to delivering highly accurate, document-specific responses. Often, these specialised documents are either too scarce or proprietary material that is not available during the pre-training stage. Hence, even the state-of-the-art instruct LLMs struggle to answer queries that require access to these documents.

A common solution is Retrieval-Augmented Generation (RAG) (Lewis et al., 2020; Karpukhin et al., 2020), which conditions LLM's responses on relevant passages retrieved from the target documents. While effective, RAG is highly sensitive to retrieval quality, and retriever failures often lead to hallucinations or incomplete answers (Ji et al. (2023); Nandwani et al. (2023)). Ingesting the new knowledge from the specialized documents into the parameters of the model can potentially alleviate the issues caused by retriever failures, as the model can fall back on its parameteric knowledge.

Unsupervised Continued Pre-Training (CPT) Ke et al. (2023) is an effective way of ingesting the new knowledge, but doing so on the Instruct LLMs results in catastrophic forgetting of the skills acquired during IFT Ke et al. (2025). As a result, most of the prior works (Ma et al., 2023; Yang et al., 2024; Lu et al., 2025) apply CPT on the 'base LLM' but have to redo IFT to re-acquire the skills present in instruct LLM. This may not always be feasible due to the lack of the IFT dataset used to create the instruct LLM.

Another way of knowledge ingestion involves direct finetuning of the instruct LLM using IFT-style training data, such as question-answers (QAs) from the new documents. However, QAs from the new documents are often not readily available and hence works such as Zhang et al. (2024b); Bhushan et al. (2025) resort to QAs generated synthetically by prompting a stronger LLM. A major advantage of such techniques is that they don't require expensive IFT after knowledge ingestion. However, there are two main issues: (1) we need to generate a massive amount of synthetic data, which may become prohibitively expensive (Yang et al., 2025), and (2) unlike CPT, it is difficult to guarantee coverage of the entire knowledge via QAs.

In this work, we ask a research question – how can we capture the efficiency of CPT to ingest new knowledge and, at the same time, retain the instruction following ability of the instruct LLMs? To tackle this challenge, we resort to the idea of editing models via task arithmetic (Ilharco et al., 2023). In task arithmetic, subtracting the base model's weights from a fine-tuned model yields the corresponding 'task vector'. This task vector captures the skills present in the fine-tuned model. One can then combine different 'task vectors' through arithmetic operations, such as addition, to impart the corresponding skills to the model. In our case, we wish to impart new knowledge of the corpus as well as the instruction following ability of the instruct LLM to the base LLM. The 'instruction following vector' of the instruct LLM, which can be easily computed by subtracting the publicly available instruct and base LLM's weights, captures the instruction following skills. To obtain a 'knowledge vector' that captures the new knowledge, we propose to train a low-rank adapter (LoRA) via CPT on top of the base LLM. Adding the 'instruct following vector' and the 'knowledge vector' to the base LLM gives us a model that has the new knowledge from the documents as well as the instruction following skills of the Instruct LLM. (see fig. 1).

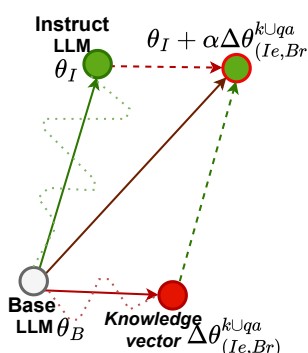

Figure 1: Exploiting task-arithmetic to combine new knowledge adapter with instruction following capabilities of Instruct LLM.

We call our method KNITLM: KNowledge IngesTion via LoRA Merging. It imparts new knowledge to an Instruct LLM via a LoRA adapter that encodes the new knowledge. Instead of training on top of instruct LLM, the 'knowledge LoRA' is trained via CPT on top of the corresponding base LLM. During inference, the knowledge LoRA is plugged with the instruct LLM, combining the knowledge of new documents in the LoRA weights with the instruction following ability of Instruct LLM to respond to document-specific queries. Note that we use the idea of task-arithmetic to motivate our method. However, in practice, we need not do any model merging, as the knowledge LoRA can be plugged directly on top of the Instruct LLM.

To enhance the recall of newly learnt knowledge, we may use a small amount of synthetic QA during training of the knowledge LoRA adapter (Allen-Zhu & Li, 2024). We note that instruct LLMs typically have additional tokens in their vocabulary, such as '[INST]' for Mistral (Jiang et al., 2023; Mistral AI; 2024). LoRAs trained on top of base LLMs would be oblivious to such tokens. To enhance the adaptability of the knowledge adapter with Instruct LLM, we replace the token embeddings in the base LLM with those from instruct LLM during training. Our ablation study shows that this improves the overall performance.

To the best of our knowledge, KNITLM is the first work that proposes an efficient way of imparting new knowledge to an existing Instruct LLM via a knowledge LoRA trained using the corresponding base LLM. To summarize, our contributions are as follows:

1. We propose KNITLM (KNowledge IngesTion via LoRA Merging), a lightweight method that efficiently ingests knowledge from new documents by first training a knowledge adaptor on the base LLM using CPT and then transferring it to the instruct LLM. This enables knowledge ingestion without the costly IFT phase following CPT.

2. To make KNITLM work, we devise a novel method that uses token embeddings from the instruct LLM during CPT on the base LLM. This significantly improves the adaptability of the knowledge adapter.

3. Through extensive experiments and ablations, we demonstrate that our method outperforms state-of-the-art SFT based knowledge infusion methods (Zhang et al., 2024b; Bhushan et al., 2025) while requiring substantially less synthetic data.

## 2 RELATED WORK

A central challenge in adapting LLMs to new specialized domains is injecting corpus knowledge while preserving general reasoning and instruction-following skills. A recent survey Song et al. (2025) categorizes ingestion methods into four categories: (1) dynamic retrieval-based approaches, (2) static knowledge injection into the parameters, (3) modular adapters, and (4) prompt optimization, outlining trade-offs between cost, adaptability, and robustness.

Prompt optimization focuses on designing effective prompts to improve domain-specific responses from the model's pre-existing knowledge without any further fine-tuning or architectural changes (Singhal et al., 2023; Yao et al., 2023). Hybrid frameworks like *DALK* further combine dynamic knowledge injection with prompt optimization for clinical domains Li et al. (2024). While promising, this paradigm suffers from the inherent limitation of in-context learning: *context window size*. Designing effective prompts is inefficient, making it unsuitable for scalable knowledge integration.

Dynamic approaches such as Retrieval-Augmented Generation (RAG) (Lewis et al., 2020; Guu et al., 2020; Karpukhin et al., 2020) ingests external knowledge in the context at inference time to ground responses on retrieved passages. Recent progress has showcased its effectiveness across diverse domains Asai et al., 2024; Qiu et al., 2023; Kim et al., 2024; Tang et al.; Yan et al., 2024. Recent refinements include joint retriever-generator training for better domain fit (Sachan et al., 2021; Siriwardhana et al., 2023; Shi et al., 2024). However, RAG-based approaches remain vulnerable to retrieval failures, leading to hallucinations (Nandwani et al., 2023; Ji et al., 2023; Setty et al., 2024).

Another research direction has been static knowledge injection, which infuses domain knowledge into the model's parameters via fine-tuning or continued pre-training (CPT), enabling closed-book inference without access to external documents (Gururangan et al., 2020; Ke et al., 2023; Wu et al., 2024; Lu et al., 2025). Various works have established the utility of CPT across multiple fields, such as medical (Wu et al., 2024; Christophe et al., 2024), materials (Zhao et al., 2025; Zhang et al., 2024a), finance (Wu et al., 2023; Shah et al., 2022; Xie et al., 2023), and education (Dan et al., 2023). Çağatay Yıldız et al. (2025) evaluates CPT across 159 domains and confirms benefits for smaller models, but notes saturation in learning at larger scales. However, static methods face scalability challenges, and direct CPT on instruction-tuned models causes regression in general skills and instruction following (Ke et al., 2025), supporting our decision to do CPT on base models instead. Several studies prove the importance of corpus format for effective CPT (Xie et al., 2024). For instance, Allen-Zhu & Li (2024) demonstrate that incorporating QA-style data during pre-training enhances knowledge extraction over mere memorization. Similarly, Mix-CPT (Jiang et al., 2025) shows that mixing instruction-style corpora during CPT enhances learnability in the post-hoc SFT. Other hybrid pipelines such as PIT, CPT+SFT, or CPT+RL can boost closed-book performance (Jiang et al., 2024; He et al., 2025; Ovadia et al., 2025), but rely on costly instruction data or heavy SFT after knowledge ingestion, which is a common key limitation for these approaches.

Modular adapters tackle scalability by training small, pluggable modules like LoRA to store domain knowledge while freezing the base model. This parameter-efficient fine-tuning (PEFT) enables adapters for diverse domains (Xu et al. (2024); Zhang et al. (2023)). While effective, both static and modular approaches typically evaluate against base LLMs on domain benchmarks, but neglect RAG as a stronger baseline. Though more scalable than static ingestion, this still depends on high-quality instruction data after knowledge ingestion to recover instruction-following abilities.

Recent works like RAFT (Zhang et al., 2024b) and PA-RAG (Bhushan et al., 2025) bridge static and dynamic paradigms. They finetune the Instruct LLM to ingest the document knowledge into the model's parameters, and use retrieved passages during inference. Despite improvements, they still heavily rely on large synthetic QA corpora.

## 3 METHODOLOGY

Let $\theta$ represent the parameters of an LLM that assigns a probability $\mathbf{Pr}(\mathbf{p}; \theta)$ to a sequence of tokens $\mathbf{p} = (t_1 \cdots t_m)$. Let $\theta_B$ and $\theta_I$ be the weights of the corresponding base and instruct LLMs, respectively. Further, let $\mathcal{D}_k = \{\mathbf{p}^i\}_{i=1}^N$ represent the new knowledge that we wish to ingest on top of the instruct LLM.

We propose to ingest the knowledge from $\mathcal{D}_k$ into a Low Rank Adapter (LoRA). A naïve way to train such an adapter would be to start with the most optimal weights $\theta_I$ and minimize the negative log likelihood over $\mathcal{D}_k$:

$$\Delta\theta_I^k = \arg\min_{\Delta\theta} \sum_{\mathbf{p}\in\mathcal{D}_k} -\log\mathbf{Pr}(\mathbf{p}; (\theta_I + \Delta\theta)) \tag{1}$$

However, Ke et al. (2025) observe that such an adapter results in deterioration of instruction following abilities of the instruct LLM $\theta_I$. This can be attributed to the fact that the instruct LLM is obtained via supervised finetuning of the base LLM, whereas the training objective in eq. (1) is unsupervised next token prediction. Next, we observe that this objective is the same as the training objective of the base LLMs. Therefore, $\theta_B$ could be more amenable to continual pretraining and thus may provide an ideal starting point for ingesting new knowledge. Motivated by this observation, we propose to train the knowledge adapter on top of the base LLM -

$$\Delta\theta_B^k = \arg\min_{\Delta\theta} \sum_{\mathbf{p}\in\mathcal{D}_k} -\log\mathbf{Pr}(\mathbf{p}; (\theta_B + \Delta\theta)) \tag{2}$$

In our notation, the superscript captures the training data and the subscript captures the starting point, *i.e.*, model initialisation. Note that the final knowledge-infused parameters returned by eq. (2) are $\theta_B + \Delta\theta_B^k$. However, such a model lacks the instruction following ability of the instruct LLM. But notice that the new knowledge from $\mathcal{D}_k$ is mainly captured in $\Delta\theta_B^k$, which may be combined with $\theta_I$ that already has instruction following abilities. Therefore, instead of using $\theta_B + \Delta\theta_B^k$ as our final parameters, we propose to use $\theta_I + \alpha\Delta\theta_B^k$, where $\alpha \in (0, 1]$ is a hyperparameter -

$$\theta^* = \theta_I + \alpha\Delta\theta_B^k \tag{3}$$

**Task-arithmetic inspired interpretation**

Ilharco et al. (2023) show that if $\theta_1$ and $\theta_2$ are two different models finetuned from the same base model $\theta_B$, then the corresponding task vectors, $\Delta\theta_B^1 = \theta_1 - \theta_B$ and $\Delta\theta_B^2 = \theta_2 - \theta_B$, capture the skills infused in them. If we combine the two task vectors, we get a model that possibly possesses both the skills -

$$\theta^c = \theta_B + \alpha\Delta\theta_B^1 + \gamma\Delta\theta_B^2 \tag{4}$$

Here, $\theta^c$ is the combined model capturing the skills of both the finetuned models. $\alpha$ and $\gamma$ are hyperparameters.

Now, let $\mathcal{D}_s$ be the data used for the instruction tuning of the instruct LLM. Then the corresponding '*instruct task vector*' capturing all the skills would be -

$$\Delta\theta_B^s = \theta_I - \theta_B = \arg\min_{\Delta\theta} \sum_{(\mathbf{x},\mathbf{y})\in\mathcal{D}_s} -\log\mathbf{Pr}(\mathbf{y}|\mathbf{x}; (\theta_B + \Delta\theta)) \tag{5}$$

We can think of our knowledge adapter $\Delta\theta_B^k$ as '*knowledge task vector*' capturing all the knowledge from the corpus. Now, combining the '*instruct task vector*' with '*knowledge task vector*', we get -

$$\theta^* = \theta_B + \alpha\Delta\theta_B^k + \gamma\Delta\theta_B^s \tag{6}$$

Substituting $\gamma = 1$ and $\Delta\theta_B^s = \theta_I - \theta_B$ from eq. (5), we get -

$$\theta^* = \theta_B + \alpha\Delta\theta_B^k + \theta_I - \theta_B = \theta_I + \alpha\Delta\theta_B^k \tag{7}$$

which is exactly same as eq. (3).

---

**Algorithm 1** KNITLM: Knowledge Adapter Training

1 **Input:** Base $(\theta_{Be}, \theta_{Br})$, Instruct $(\theta_{Ie}, \theta_{Ir})$, Corpus $\mathcal{D}_k$, QA set $\mathcal{D}_{qa}$, learning rate $\eta$, epochs $T$
2 **Output:** Knowledge LoRA $\Delta\theta^{k\cup qa}_{(Ie,Br)}$

3 **Model Init.:** $\theta \leftarrow (\theta_{Ie}, \theta_{Br})$
4 **LoRA Init.:** $\Delta\theta \leftarrow (\mathbf{0}, \Delta\theta_{(Ie,Br)})$
5 **Union data:** $\mathcal{D}_{k\cup qa} \leftarrow \mathcal{D}_k \cup \mathcal{D}_{qa}$
6 **for** $t = 1$ **to** $T$ **do**
7    **for** *mini-batch* $\mathcal{B} \subset \mathcal{D}_{k\cup qa}$ **do**
      // Compute loss as in eq. (9)
      $\mathcal{L}_\mathcal{B} \leftarrow \sum_{\mathbf{x}\in\mathcal{B}} -\log \mathbf{Pr}\left(\mathbf{x}; \left(\theta_{Ie}, \theta_{Br} + \Delta\theta_{(Ie,Br)}\right)\right)$

      $\Delta\theta_{(Ie,Br)} \leftarrow \Delta\theta_{(Ie,Br)} - \eta\nabla\mathcal{L}_\mathcal{B}$
   **end**
**end**
8 Set $\Delta\theta^{k\cup qa}_{(Ie,Br)} \leftarrow \Delta\theta_{(Ie,Br)}$
9 **return** $\Delta\theta^{k\cup qa}_{(Ie,Br)}$

---

**Adding a small amount of synthetic QA :** Allen-Zhu & Li (2024) observe that having a few question-answer pairs in the pre-training data significantly enhances the recall of the ingested knowledge. These QAs do not have to necessarily span the entire corpus.

Accordingly, we enhance our training corpus with a small amount of synthetically generated QAs. Unlike instruction fine-tuning, where loss is backpropagated only over the answer tokens conditioned on the question, we concatenate the question, answer and treat it as part of the new knowledge to be ingested. Accordingly, let $\mathcal{D}_{qa} = \{\mathbf{x}^i = (\mathbf{sys}, \mathbf{q}^i, \mathbf{a}^i)\}_{i=1}^{nq}$ be the training data obtained from the synthetic QAs. Here **sys** is a common system prompt that asks the model to answer the question; $(\mathbf{sys}, \mathbf{q}^i, \mathbf{a}^i)$ represents the concatenation of system prompt, question and answer; and $nq$ is the number of synthetically generated QAs. We train our knowledge adapter on top of the base LLM using $\mathcal{D}_{k\cup qa} = \mathcal{D}_k \bigcup \mathcal{D}_{qa}$.

$$\Delta\theta^{k\cup qa}_{B} = \arg\min_{\Delta\theta} \sum_{\mathbf{x}\in\mathcal{D}_k \bigcup \mathcal{D}_{qa}} -\log \mathbf{Pr}(\mathbf{x}; (\theta_B + \Delta\theta)) \tag{8}$$

**Using token embeddings of the instruct LLMs**

We observe that for certain tokens, embeddings in the base and instruct LLMs are quite different. Often, they correspond to the tokens introduced during the instruction fine-tuning phase, *e.g.*, '[INST]' in instruct versions of Mistral. While training the low-rank knowledge adapter, we often target the linear layers and do not fine-tune the token embeddings, *i.e.*, $\Delta\theta$ typically contains non-zero entries only for attention and MLP layers. In addition, the system prompt used in the synthetic QA dataset also introduces some of these tokens not seen during training of the base LLM, *i.e.*, parameters $\theta_B$ are oblivious to these tokens. This creates a mismatch: the knowledge adapter is trained with the base model's token embeddings but during inference it is used with the instruct LLM's entirely different embeddings. To mitigate this, we propose to use the token embeddings of the instruct LLM instead of the base LLM. Concretely, during knowledge LoRA training, we replace the base LLM's token embeddings (and the `lm_head`, if separate) with those of the instruct LLM. We claim that this enhances the adaptability of the knowledge LoRA, trained on the base LLM but used with an instruct LLM. Intuitively, it gives the adapter parameters early exposure to the inference-time environment and vocabulary, reducing the risk of distribution shift.

If we represent model parameters $\theta_B$ as $(\theta_{Be}, \theta_{Br})$ and $\theta_I$ as $(\theta_{Ie}, \theta_{Ir})$, where $\theta_{Be}, \theta_{Ie}$ are the token embeddings and $\theta_{Br}, \theta_{Ir}$ are the remaining parameters in the base and instruct LLMs, respectively, then we learn our knowledge LoRA on top of $(\theta_{Ie}, \theta_{Br})$ -

$$\Delta\theta^{k\cup qa}_{(Ie,Br)} = \arg\min_{\Delta\theta} \sum_{\mathbf{x}\in\mathcal{D}_k \bigcup \mathcal{D}_{qa}} -\log \mathbf{Pr}\left(\mathbf{x}; (\theta_{Ie}, \theta_{Br} + \Delta\theta)\right) \tag{9}$$

Algorithm 1 presents our method that returns the trained knowledge LoRA. One can load it on top of the instruct LLM $\theta_I$ to obtain the final model parameters as -

$$\theta^* = \theta_I + \alpha\left(\mathbf{0}, \Delta\theta^{k\cup qa}_{(Ie,Br)}\right) = (\theta_{Ie}, \theta_{Ir}) + \alpha\left(\mathbf{0}, \Delta\theta^{k\cup qa}_{(Ie,Br)}\right) = \left(\theta_{Ie}, \theta_{Ir} + \alpha\Delta\theta^{k\cup qa}_{(Ie,Br)}\right) \tag{10}$$

## 4 EXPERIMENTAL SETUP

We seek to answer the following research questions through our experiments:

1. Can KNITLM effectively ingest knowledge into instruct LLM's parameters? To test this, we evaluate the knowledge ingested model in the QA setup where it is provided with only the question and it has to answer from its parametric knowledge.

2. Can KNITLM effectively combine the knowledge ingested in its parameters with additional knowledge present in its context? To test this, we compare KNITLM in the RAG setup with SFT based methods such as RAFT (Zhang et al., 2024b) and PA-RAG (Bhushan et al., 2025) that rely heavily on an enormous amount of synthetic data.

3. What is the role of synthetic QA data in KNITLM? Specifically, is our method robust to the size of the synthetic QA dataset and its coverage of the corpus? To this end, we run two ablations – (1) We vary the size of the synthetic dataset and compare KNITLM with RAFT and PARAG in both QA and RAG setups. (2) We systematically bias the synthetic QA dataset by generating training QAs from a specific subset of documents and then measure the impact on performance.

4. Finally, we seek to quantify the importance of using token embeddings of the instruct LLM instead of base LLM while training the knowledge adapter, *i.e.*, what happens if we train the adapter on $\theta_B = (\theta_{Be}, \theta_{Br})$ (eq. (8)) instead of $(\theta_{Ie}, \theta_{Br})$ (eq. (9)).

### 4.1 Datasets, Models, Evaluation Metrics and Training Details

**Datasets:** We train all our models on two datasets introduced in Bhushan et al. (2025). Both datasets consist of text from a technical Redbook[1] along with corresponding test question answers To test the performance in the RAG setup, they also provide a list of retrieved passages for each question.

While manually inspecting the test data, we observed that some question-answers in the original test set are either incomplete or not properly decontextualized. Therefore, we decided to clean up the test data by prompting Llama-3.1-70B-Instruct to evaluate each QA pair on various dimensions and assign a rating from 1 to 10. We filtered all QA pairs with a score less than 10. The resulting datasets have 313 and 1554 test samples, dropping 26% and 32% of the QAs in the original version. Our small-scale human study reveals that our LLM filter is able to recall 70% of the improper QAs from the test data, thereby improving its quality. See appendix A for the details of the human study and appendix F for the prompt used for cleaning the test data.

**Models:** We ingest the knowledge from the books into *Mistral-7B-Instruct-v0.3*[2] and *LLama-3.1-8B-Instruct*[3]. These models are selected such that their training cutoff predates the publication of both Redbooks, ensuring that the models would not have seen these documents during their pretraining.

**Evaluation Metrics:** We evaluate our models in two setups – QA and RAG. In the QA setup, the model is prompted with only the question, and in the RAG setup, we provide the top 5 retrieved passages along with the question. We use Llama-3.3-70B-Instruct as a judge to evaluate the correctness of the predicted answer *w.r.t.* the given gold answer. For each test sample, we provide the judge with the question, gold answer, and generated answer, and the judge returns a binary score (0/1) after reasoning across multiple criteria. Full prompt details are in appendix F.

To ensure that our LLM judge is aligned with human judgement, we conduct a small-scale human study in which we evaluate the responses generated by the instruct LLM. Our LLM judge exhibits ∼86% and ∼97% agreement with humans in the QA and RAG setup. It is interesting to note the difference in the agreement rate of the two setups. We attribute this difference to the fact that in the QA setup, instruct LLM's responses are not grounded on any text. The model's responses generated solely from its parameteric memory tend to be more verbose, confusing the LLM and human judges alike. In fact, we observe that in the QA setup, inter-annotator agreement amongst humans is also lower than that in the RAG setup. See appendix A for more details on the human study.

**Baselines:** We compare KNITLM with RAFT Zhang et al. (2024b) and PA-RAG Bhushan et al. (2025). Both baselines rely on synthetically generated QAs for knowledge ingestion. We prompt

---

[1] Book 1: Do More with Less: Automating IBM Storage FlashSystem Tasks with REST APIs, Scripting, and Ansible. Book 2: Red Hat OpenShift Container Platform on IBM Z and LinuxONE.

[2] mistralai/Mistral-7B-Instruct-v0.3

[3] meta-llama/LLama-3.1-8B-Instruct

| | | Book 1 | | | | | Book 2 | | | |
|---|---|---|---|---|---|---|---|---|---|---|
| | QA | RAG | | | Train Time (in mins) | QA | RAG | | | Train Time (in mins) |
| | | All | Ret. Success | Ret. Fail. | | | All | Ret. Success | Ret. Fail. | |
| Instruct | 53.67 | 71.76 | 86.27 | 54.17 | | 27.51 | 61.23 | 77.96 | 36.13 | |
| RAFT | 56.87 | 79.87 | 88.20 | 68.89 | 19 | 27.23 | 62.95 | 79.16 | 38.65 | 35 |
| PA-RAG | 64.22 | 84.66 | **92.70** | 74.07 | 43 | 27.23 | 62.23 | 78.60 | 37.64 | 62 |
| KNITLM | **73.80** | **86.58** | 92.13 | **79.26** | 7 | **40.98** | **66.86** | **80.67** | **46.13** | 13 |

Table 1: Comparing KNITLM with various baselines. The table reports the fraction of test samples where the LLM Judge rated the predicted response as good as the gold response. **QA**: performance in the QA setup; **All**: performance over the entire test set in RAG setup; **Ret. Success**: performance over test queries where retriever succeeds (match@5=1); **Ret. Fail.**: performance over test queries where retriever fails (match@5=0). These results are for Mistral-7B-Instruct-v0.3. Please see Table 9 for results on Llama-3.1-8B-Instruct. **Train Time**: Approximate training time in minutes.

Mixtral-8x22B-Instruct-v0.1 to generate synthetic QAs and use the same prompt as described in Bhushan et al. (2025). See appendix F for the exact prompt.

**Size of the synthetic training data:** The number of question–answer pairs in the synthetically generated training dataset depends on the corpus size. For RAFT and PA-RAG we need to cover the entire corpus with the generated synthetic data. Therefore, we generate pairs such that the total number of generated words is twice the number of words in the corpus. PA-RAG additionally requires multiple answers per question. For each training question, we generate four additional answers using Mixtral-8x22B-Instruct-v0.1. Consequently, the synthetic dataset for PA-RAG contains about $\sim 10$ times as many words as the corpus. See appendix B for the exact sizes of the training and test datasets.

Recall that KNITLM also requires a small amount of synthetic QAs, but without the need to cover the full corpus. Consequently, for KNITLM, we randomly select question-answer pairs such that the total number of selected words is only 50% of the number of words in the corpus.

**Training Details:** We run all experiments with Hugging Face's `SFTTrainer`[4]. We applied LoRA to *all* linear layers in the model with rank $r = 16$. For KNITLM, after loading the base model, we replace its token embeddings (and `lm_head` if they are separate) with those of the corresponding instruct model as explained in section 3.

For baseline methods, model selection is based on validation loss with early stopping. For KNITLM we instead train until convergence of the training loss and control overfitting via the scaling hyperparameter $\alpha$. We sweep $\alpha \in \{0.25, 0.5, 0.75, 1.0\}$ and select the best value using validation performance. Notably, applying early stopping to KNITLM may yield additional gains, as observed in appendix D.

## 5 EXPERIMENTAL RESULTS

### 5.1 COMPARISON WITH THE BASELINES

We compare KNITLM with the corresponding instruct LLM, RAFT and PA-RAG in table 1 for both the datasets in both the QA and RAG setups. In addition to compute the performance over the entire test set in the RAG setup, we further split the performance depending on the retriever's success in fetching the gold passages. Recall that KNITLM utilizes only a fraction of the synthetic QAs required by PA-RAG. Despite such a low utilizaton, KNITLM outperforms both RAFT and PA-RAG in both QA and RAG setups.

For Book1, performance of KNITLM in the QA setup (73.8) is better than the Instruct LLM in the RAG setup (71.76). This gain can be attributed to successful knowledge ingestion by KNITLM.

---

[4]docs/trl/sft_trainer

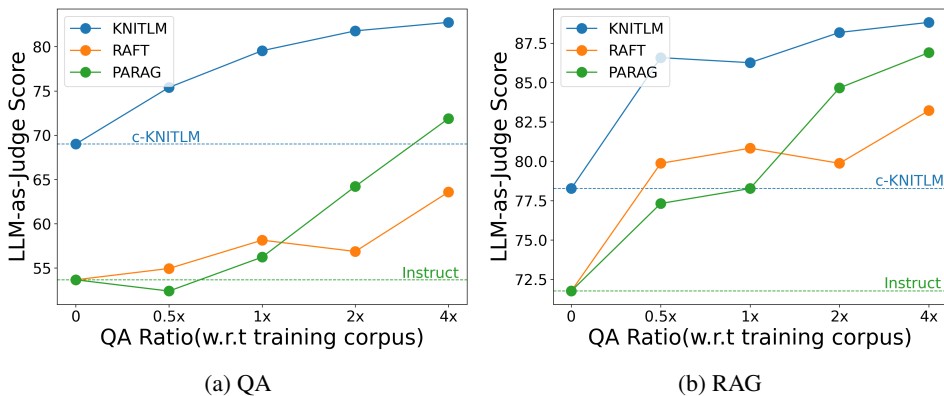

(a) QA                                (b) RAG

Figure 2: Impact of scaling synthetic QA on the Redbook1 dataset with Mistral-Instruct-v0.3. Blue horizontal line corresponds to c-KNITLM– our model trained in an unsupervised manner only using the document text. Green horizontal line corresponds to Mistral-Instruct-v0.3.

However, note that the performance of KNITLM in the QA setup (73.8) is not at par with the instruct LLM when the retriever is able to fetch the gold passage (86.27), indicating that retrieved passages can still improve the performance of KNITLM. When provided with the retrieved passages, KNITLM is successfully able to exploit them when the retriever succeeds, as its performance jumps to 92.13. When the retriever fails, KNITLM is able to ignore the context and answer using the knowledge infused in model's parameters. One might expect performance with only distractor passages (retriever failure) to be lower than in the QA setup (no distractors). Surprisingly, it is not the case (79.26 vs. 73.8). We hypothesize that even distractor passages in the context help knowledge-infused models to recall relevant information from their parameters. This observation holds across RAFT, PA-RAG, and KNITLM. We observe similar trends for Book2.

## 5.2    IMPACT OF THE SIZE OF THE SYNTHETIC DATA

In this experiment, we study how scaling the synthetic QA dataset affects model performance. For this, we progressively increase the number of synthetic QA pairs and compare KNITLM with baselines in both QA and RAG setups. We define '*QA ratio*' as the ratio of number of words in the synthetic QA dataset to the words in the training document. For KNITLM, QA ratio of 0 corresponds to training only on the corpus (eq. (3)) and we call it corpus-KNITLM or c-KNITLM in short. For RAFT and PA-RAG , 0 corresponds to the instruct model. Note that for PA-RAG , we need to generate multiple answers for each question, and the QA ratio does not account for it. Therefore, the actual synthetic QA dataset used for PA-RAG would contain about $5\times$ as many words as RAFT and KNITLM. Note that the performance numbers in our main experiments correspond to a ratio of 2 for RAFT and PA-RAG , and 0.5 for KNITLM. For this analysis, we generate additional data and scale up to a ratio of 4.

Figures 2a-2b presents the analysis. We first observe that c-KNITLM(dotted blue horizontal line) outperforms both baselines in the QA setup, demonstrating the capability of our method to efficiently ingest knowledge.

As seen in Figure-2b KNITLM can achieve near optimal performance with only $0.5\times$ of synthetic data where the difference of performance is only 2.24% between $0.5\times$ vs $4\times$ of synthetic data. In contrast, PA-RAG improves by more than 10% going from $0.5\times$ to $4\times$ synthetic data, implying PA-RAG indeed needs comprehensive volume of synthetic data for effective knowledge ingestion. A similar trend appears in the QA setup, where KNITLM not only outperforms the baselines but also shows a smoother saturation curve, unlike the sharp jumps with more synthetic data as seen in PA-RAG. These results empirically establish that KNITLM is indeed much more lightweight yet the new state-of-the-art scalable knowledge ingestion recipe, which does not need extensive synthetic data generation.

### 5.3 ROBUSTNESS OF KNITLM TO CORPUS COVERAGE BY SYNTHETIC QAS

In the previous experiment, we observed that KNITLM is robust to the amount of synthetic QA dataset and its RAG performance begins to saturate even with QA ratio of 0.5. Here, we systematically study the impact of partial knowledge coverage on our method. Allen-Zhu & Li (2024) note that "*partially augmenting data can improve knowledge extraction for non-augmented data*". Here augmentation refers to adding QAs corresponding to the knowledge being ingested. To systematically study this, we run a control experiment – we train a version of KNITLM using the document along with QA data generated from only chapters 1 to 3 of Book1 (we call it biased-KNITLM, or

|  | Ch. 1-3 | | Ch. 4-5 | |
|---|---|---|---|---|
|  | **QA** | **RAG** | **QA** | **RAG** |
| c-KNITLM | 52.80 | 74.40 | 65.96 | 80.85 |
| b-KNITLM | +26.40 | +8.00 | + 2.66 | +5.25 |

Table 2: Comparison between models trained without synthetic QA (c-KNITLM i.e., corpus-KNITLM) and models trained with chapter-biased synthetic QA(b-KNITLM i.e., biased-KNITLM). Reported are LLM-as-Judge scores for QA and RAG setup for two chapter splits.

b-KNITLM) and compare its performance with c-KNITLM (trained without any QA data). Table 2 shows the results. We observe that adding synthetic QAs from chapters 1 to 3 improves the performance even on chapter 4-5. Our result supports the observation in Allen-Zhu & Li, implying that even if we have access to QA from only a part of the corpus, KNITLM will still show gains over the remaining data.

### 5.4 IMPACT OF USING INSTRUCT LLM'S TOKEN EMBEDDINGS DURING TRAINING

Recall that in KNITLM we replace the frozen token embeddings $\theta_{Be}$ in the base LLM with those from the corresponding instruct LLM. *I.e.*, we train the knowledge LoRA adapter on top of $(\theta_{Ie}, \theta_{Br})$ instead of $(\theta_{Be}, \theta_{Br})$. In this experiment, we quantify its impact by comparing the models trained using eq. (8) and eq. (9), respectively. Table 3 shows the results. For easy reference, we copy the result of PARAG from Table 1.

We find that using instruct LLM's token embeddings improves performance in both QA and RAG setups. Without them, performance drops significantly under retriever failure cases and approaches that of PA-RAG. Thus, replacing the base model's embeddings with those of the instruct model is crucial for KNITLM to outperform PA-RAG in the RAG setup.

Overall, this ablation confirms that using instruct token embeddings is a simple yet effective intervention: it resolves the vocabulary mismatch between the base and instruct LLMs, thereby improving the adaptability of knowledge adapters during inference.

|  | QA | RAG | | |
|---|---|---|---|---|
|  |  | **All** | **Ret. Success** | **Ret. Failure** |
| PA-RAG | 64.22 | 84.66 | 92.70 | 74.07 |
| KNITLM | 73.80 | 86.58 | 92.13 | 79.26 |
| e-KNITLM | 72.76 | 84.57 | 92.13 | 73.13 |

Table 3: Effectiveness of using instruct LLM's embeddings during training. Comparing KNITLM with a version trained directly on top of base LLM (e-KNITLM) for Book 1.

## 6 CONCLUSION

In this work, we introduced KNITLM, a lightweight and efficient approach for knowledge infusion in pre-trained LLMs. By training a knowledge adapter through CPT on the base LLM and transferring it to the instruct LLM, KNITLM enables effective knowledge ingestion without the costly IFT phase. To further enhance adaptability, we leverage token embeddings from the instruct LLM during CPT, thereby strengthening the transferability of the knowledge adapter.

Our experiments and ablation study show that KNITLM not only achieves superior performance in RAG setups but also consistently outperforms state-of-the-art SFT-based knowledge infusion methods, such as RAFT and PA-RAG, while requiring substantially less synthetic data. These results highlight KNITLM as a practical and scalable alternative for rapidly incorporating domain-specific knowledge into LLMs.

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

# A  HUMAN ANNOTATION AND LLM-AS-A-JUDGE ALIGNMENT

## A.1  HUMAN ANNOTATION SETUP

To validate the reliability of our evaluation protocol, we conduct a human annotation study using 50 examples sampled from the Book 1 and Book 2 test splits. Responses were generated with *Mistral v0.3 Instruct* under both QA and RAG setups. For each instance, annotators were provided with the question, the gold answer, and the model generated answer. Each example was independently rated by three domain experts according to the rubric below:

- **Fully Correct (1):** Response covers all statements in the gold, introduces no contradictions, and may include additional relevant information.
- **Incorrect (0):** Response contradicts the gold, fails to answer the question, or is incomplete/vague.
- **Ill-formed QA (–1):** The question or gold answer is itself vague, incomplete, or not properly decontextualized.

In cases where all three annotators disagreed, a fourth expert adjudicated to obtain the final label. The final human score was determined via majority vote.

## A.2  HUMAN ANNOTATION RESULTS

Annotation statistics are shown in Table 4. We annotate 50 model responses for both QA and RAG setups in Book 2, and an additional 50 responses for the QA setup on Book 1, since scores of 0 were over-represented in the QA annotations of Book 2. Inter-annotator agreement is strong for the RAG setup, with consistently high percent agreement and Krippendorff's $\alpha$ values, reflecting stable human judgments. The QA setup shows a lower agreement ($\alpha \approx 0.66$ compared to $\approx 0.92$ for RAG), which we attribute to the longer and more verbose responses (196 words on average vs. 135 in RAG). These longer responses often include hallucinations or extraneous details, making annotation more challenging.

Table 4: Human annotation agreement statistics

| Setup | Agreement | Krippendorff's $\alpha$ | AC2 | Annotators | Examples | Response Word Count |
|-------|-----------|------------------------|------|-----------|----------|--------------------|
| QA | 0.78 | 0.66 | 0.68 | 3 | 100 | 196 |
| RAG | 0.95 | 0.92 | 0.92 | 3 | 50 | 135 |

During annotation, a notable fraction of examples were identified as Ill-formed QA pairs, reflecting limitations of the synthetic test sets (Table 5).

Table 5: Filtered examples by humans

| Dataset | Ill-formed | Valid | Total |
|---------|-----------|-------|-------|
| Book 1 | 9 | 41 | 50 |
| Book 2 | 15 | 35 | 50 |

## A.3  LLM-AS-A-JUDGE FOR FILTERING

To mitigate dataset noise, we employ *Llama 3.1 70B Instruct* as an automatic judge. Each evaluation instance provided the judge with the question and gold answer, and the judge assigns a rating (1–10) based on Accuracy, Relevance, Clarity, and Usefulness (see appendix F for the prompt). QA pairs with ratings $< 10$ were filtered out. We adapted our prompt from Synthetic Data Kit

This automatic filtering removes $\sim 71\%$ of the Ill-formed QA pairs identified by humans. Extending this procedure to the full test dataset yields the results in Table 6.

Examples of removed QA pairs are provided in appendix F.

Table 6: Dataset size before and after filtering

| Dataset | Before | After |
|---------|--------|-------|
| Book 1  | 425    | 313   |
| Book 2  | 2269   | 1554  |

### A.4 LLM-AS-A-JUDGE FOR EVALUATION

We use *Llama 3.3 70B Instruct* as the LLM-as-a-Judge to evaluate the generated responses for all of our experiments. To verify its reliability, we compared the judge's binary decisions (0/1) against the human majority labels on the annotated examples after filtering. The results, shown in Table 7, demonstrate a strong alignment between the LLM-as-a-Judge and human judgments, indicating that the prompt (detailed in appendix F) used produces consistent evaluations throughout the data set.

Table 7: Alignment of LLM-as-a-Judge with human annotations

| Dataset | Accuracy | Precision | Recall | TN | FP | FN | TP | Total |
|---------|----------|-----------|--------|----|----|----|----|-------|
| QA      | 0.84     | 0.86      | 0.76   | 39 | 4  | 8  | 25 | 76    |
| RAG     | 0.97     | 1.00      | 0.94   | 18 | 0  | 1  | 16 | 35    |

### A.5 DISCUSSION

Overall, the LLM-as-a-Judge demonstrates strong alignment with human annotations, achieving $\sim 84\%$ accuracy on QA and $\sim 97\%$ on RAG. The comparatively lower accuracy on QA reflects the inherent ambiguity in evaluating context-free generations, where hallucinations and verbose answers introduce annotator disagreement.

These findings suggest that (i) the synthetic test sets contain a non-trivial proportion of **Ill-formed QA pairs**, and (ii) LLM-as-a-Judge provides a reliable and scalable mechanism for filtering and evaluating examples in large-scale experiments.

## B DATA STATISTICS

Please refer to table table 8 for details about both the datasets used in the paper. As mentioned in section section 4.1, PA-RAG and RAFT train sets were created with 2x the amount of words in the domain documents.

## C RESULTS ON LLAMA

The main table with the results of KNITLM as well as various other baselines using LLaMA 3.1 8B model are presented in table table 9.

| Dataset | Chapters | Words | Train Samples PA-RAG | Train Samples RAFT | Num. Test Samples |
|---------|----------|-------|----------------------|--------------------|--------------------| 
| RedBook 1 | 5 | 15,225 | 1,107 | 286 | 313 |
| RedBook2  | 6 | 33,795 | 2,980 | 770 | 1,554 |

Table 8: Data statistics for the datasets used in the paper.

| | Book 1 | | | | Book 2 | | | |
| --- | --- | --- | --- | --- | --- | --- | --- | --- |
| | QA | RAG | | | QA | RAG | | |
| | | All | Ret. Success | Ret. Failure | | All | Ret. Success | Ret. Failure |
| Instruct | 52.40 | 67.41 | 83.71 | 45.93 | 26.61 | 59.86 | 78.97 | 31.13 |
| RAFT | 60.06 | 77.96 | 88.76 | 63.70 | 31.61 | 65.44 | 80.34 | 43.06 |
| PA-RAG | 65.81 | 82.75 | 92.70 | 69.63 | 31.87 | 64.13 | 79.03 | 41.77 |
| KNITLM | 72.20 | 80.19 | 89.89 | 67.41 | 34.92 | 64.54 | 80.58 | 40.39 |

Table 9: Main table comparing the performance of various baselines descibed in the paper using Llama 8b model.

## D    STOPPING CRITERIA ABLATION

In continual pre-training (CPT), a practical challenge is determining when to stop training. Stopping too early risks underfitting, while stopping too late may lead to overfitting to the training corpus. This decision is particularly relevant when merging the CPT base model with an instruction-tuned model. The objective of this ablation is to illustrate how the choice of stopping point affects downstream performance.

To study this effect, we conduct experiments on the Book 1 corpus by continually pre-training the *LLaMA 3.1 8B* base model for 60 epochs on KNITLM's training data mixture. At intermediate checkpoints, we perform task-arithmetic merges with the instruct model using four different merge weights (0.25–1.0) applied to the knowledge-ingested base model. At each checkpoint, we selected the optimal merge according to the LLMaJ Score under RAG setup. We conduct evaluations under both QA and RAG setups. For RAG, the Book 1 validation set was split into two subsets: **(i) Ret. Success**, where the retrieved context passages contain the answer, and **(ii) Ret. Fail**, where the context does not contain the answer.

The resulting performance trends are shown in Figures 3a-3d.

Across all setups, we observe a consistent trend: performance improves substantially in the early and mid stages of training, peaks at intermediate checkpoints, and then gradually declines as training continues to convergence. For the sake of uniformity across baselines and experimental conditions, we opted to train until convergence before performing merges. As a result, the scores presented in the main article should be viewed as *conservative estimates*. More careful stopping criteria could further enhance performance.

## E    PERFORMANCE ON GENERAL TASKS

We compare KNITLM with Llama 3.1 8B Instruct (Instruct), RAFT, and PARAG on several benchmarks:

- **Big Bench Hard** (Suzgun et al., 2022): 23 challenging tasks spanning language understanding and reasoning.
- **GPQA** (Rein et al., 2024): Google-Proof Graduate-level STEM questions.
- **MATH-Hard** (Hendrycks et al., 2021): Difficult math competition questions.
- **MMLU-Pro** (Wang et al., 2024): 12k questions across diverse fields, measuring general knowledge.
- **MUSR (Multistep Soft Reasoning)** (Sprague et al., 2024): Evaluates reasoning capabilities of LLMs.

table 10 reports the performance for Book 1.

KNITLM maintains competitive performance across all general benchmarks, while RAFT and PARAG show regression on general tasks relative to Instruct.

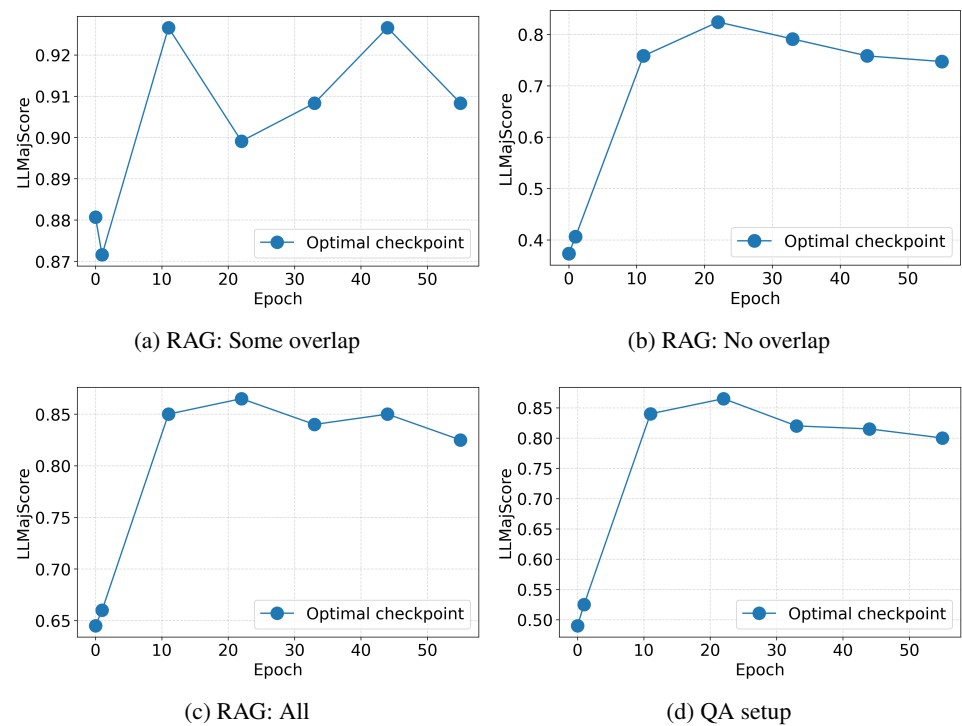

Figure 3: Stopping criteria ablation: best LLMajScore across checkpoints.

|  | **Big Bench Hard** | **GPQA** | **MATH-Hard** | **MMLU Pro** | **MUSR** | **Aggregate** |
|---|---|---|---|---|---|---|
| Instruct | 29.88 | 5.36 | 17.47 | 37.83 | 8.73 | 19.85 |
| RAFT | 29.75 | 6.22 | 14.87 | 37.67 | 6.01 | 18.90 |
| PARAG | 30.28 | 4.85 | 14.63 | 36.95 | 6.73 | 18.69 |
| KNITLM | 29.69 | 7.59 | 17.11 | 38.08 | 6.75 | 19.84 |

Table 10: General Task Performance

## F PROMPTS AND EXAMPLES

This appendix presents the prompts used for three purposes: (i) filtering low-quality QA pairs from the dataset, (ii) evaluating responses generated by LLMs, and (iii) generating synthetic QA pairs. We also provide examples of QA pairs that were removed during the filtering process, along with sample responses from our method and the baseline models.

### F.1 FILTERING PROMPT

The following prompt was used to identify Ill-formed QA pairs during dataset filtration. The filtering judge receives a question and its gold answer as input. It considers multiple criteria such as accuracy, relevance, clarity and usefulness and outputs a score from 1–10.

---

**Filtering Prompt**

```
Rate each question-answer pair on a scale from 1-10, based on:
- Accuracy (0-3): factual correctness
- Relevance (0-2): relevance to content
- Clarity (0-2): clear language
- Usefulness (0-3): value for model learning
```

```
YOU MUST RETURN A VALID JSON OBJECT OR ARRAY WITH THIS EXACT SCHEMA
    :
{{
  "question": "Exact question text",
  "answer": "Exact answer text",
  "explanation": {{
    "Accuracy": "Short explanation of factual correctness",
    "Relevance": "Short explanation of relevance",
    "Clarity": "Short explanation of clarity",
    "Usefulness": "Short explanation of usefulness"
  }},
  "Accuracy": 2,
  "Relevance": 2,
  "Clarity": 2,
  "Usefulness": 2,
  "rating": 8
}}

OR FOR MULTIPLE PAIRS:
[
  {{
    "question": "Q1",
    "answer": "A1",
    "explanation": {{
      "Accuracy": "Explanation for Accuracy",
      "Relevance": "Explanation for Relevance",
      "Clarity": "Explanation for Clarity",
      "Usefulness": "Explanation for Usefulness"
    }},
    "Accuracy": 2,
    "Relevance": 2,
    "Clarity": 2,
    "Usefulness": 2,
    "rating": 8
  }},
  {{
    "question": "Q2",
    "answer": "A2",
    "explanation": {{
      "Accuracy": "Explanation for Accuracy",
      "Relevance": "Explanation for Relevance",
      "Clarity": "Explanation for Clarity",
      "Usefulness": "Explanation for Usefulness"
    }},
    "Accuracy": 3,
    "Relevance": 2,
    "Clarity": 2,
    "Usefulness": 2,
    "rating": 9
  }}
]

*** YOUR RESPONSE MUST BE VALID JSON AND NOTHING ELSE - NO
    EXPLANATION, NO MARKDOWN ***

QA pairs to rate:
{pairs}
```

## F.2   LLM-AS-A-JUDGE PROMPT

The following prompt was used to evaluate model-generated responses. The model is provided with the question, gold answer and model generated answer, and it outputs a binary rating (0/1) according to the specified evaluation rules.

---

**LLM Evaluation Prompt**

```
You are an evaluator. Your task is to compare a Ground-truth Answer
    and a Prediction to decide if the Prediction correctly answers
    the given Question.

Evaluation Rules:
(1) Correctness: A correct prediction must include all essential
    information from the Ground-truth Answer. Extra information is
    allowed if it does not contradict the Ground-truth. If the
    Prediction states something as a possibility, treat it as a
    definitive statement.

(2) Function, Tool Names, and API Calls: If the Ground-truth Answer
    contains specific function names, tool names, API calls, or
    exact command identifiers, the Prediction must contain the same
    identifier(s) or clearly equivalent forms. Minor syntactic or
    formatting variations that do not change meaning should be
    treated as equivalent. For example, leading flag prefixes such
    as -, --, or no prefix at all when they clearly refer to the
    same option;  underscore vs hyphen differences in identifiers
    when the intent is identical;  surrounding punctuation or
    formatting differences such as backticks, quotes, parentheses,
    or code block notation; small whitespace differences or
    capitalization differences that do not change the identifier's
    meaning etc. However, replacements that change the actual
    function/tool/API name, or substitute a different command that
    would change the behavior are considered incorrect. Do not
    penalize a prediction if it contains additional function / tool
    / API names as long as the ones present in the Ground-Truth are
    covered.

(3) URLs: If the Ground-truth Answer contains specific URLs, the
    Prediction should reference the same URL or an equivalent
    canonical form. Minor differences that do not change the target
    resource (for example, presence or absence of a trailing slash,
    or http vs https when both resolve to the same canonical
    resource) should be treated as equivalent. Altering the domain,
    path, or query such that the resource is different is incorrect.

Scoring Rules:
If the Prediction is correct according to the above rules, output <
    score>1</score>. If the Prediction is incomplete or incorrect,
    output <score>0</score>.

Output Format:
<explanation>
...
</explanation>
<score>
...
</score>

First provide reasoning inside <explanation> and </explanation>
    tags. Then output the score as specified above within <score>
```

```
        and </score> tags. Do not include any extra text outside these
        tags.
```

### F.3 PROMPT FOR GENERATING SYNTHETIC QA

The prompt generates fully contextualized question–answer pairs from a document, covering the entire content and formatted with specific tags.

> **QA Generation Prompt**
>
> ```
> Create question answer pairs from the document given below within <
>     document> tags. Title of the document is given in the first line
>     of the document. Do not use co-referencing and pronouns at all
>     in the questions. Do not refer to the document in the question
>     like "according to the document ..." or any similar paraphrasing
>     . When needed, contextualize the question by using the topic
>     that the question is about. You can use the title of the
>     document as well for contextualizing. There are several figures
>     in the document, while referring to the figure in any question,
>     contextualize it by mentioning the title of the passage it was
>     present in. Put questions within <question> and </question> tags
>     and answers within <answer> and </answer> tags. Ensure that the
>     question and answers cover the entire document. When you are
>     done generating QA pairs, generate </done> token.
> ```

### F.4 EXAMPLES OF QA PAIRS REMOVED DURING FILTERING

Below are three representative examples of QA pairs that were filtered out by the LLM-as-a-Judge. Each example shows the question and gold answer.

> **Question:** What is the state of the second volume to be mapped to the hostcluster?
> **Gold Answer:** The state of the second volume to be mapped to the hostcluster is present.
>
> 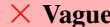 **Vague**

> **Question:** What was the status, memory, and CPU usage of the voting-app-worker-py-1 pod in the Red Hat OpenShift Cluster on IBM LinuxONE?
> **Gold Answer:** The status, memory, and CPU usage of the voting-app-worker-py-1 pod in the Red Hat OpenShift Cluster on IBM LinuxONE were as follows:
> - Status: Completed
> - Memory: Not specified
> - CPU: Not specified
>
> 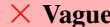 **Incomplete**

> **Question:** What are the settings for the infra nodes in the deployment of Red Hat OpenShift on IBM Z?
> **Gold Answer:** The infra nodes have settings such as disk size, RAM, vCPU, VM name, IP address, and hostname. These settings are defined in Section 10 of the all.yaml file.
>
> 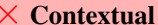 **Contextual**

### F.5 EXAMPLE MODEL RESPONSES

> **Question:** What command is used to upgrade the IBM Storage Virtualize collection to the latest version from Ansible Galaxy?
> **Gold Answer:** The command used to upgrade the IBM Storage Virtualize collection to the latest version from Ansible Galaxy is:
> *ansible-galaxy collection install ibm.storage_virtualize –force*

> To upgrade the IBM Storage Virtualize collection to the latest version from Ansible Galaxy, you can use the following command:
> "' *ansible-galaxy collection install ibm.storwize:latest* "'

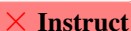

> The command used to upgrade the IBM Storage Virtualize collection to the latest version from Ansible Galaxy is:
> "'*bash ansible-galaxy collection install -r requirements.yml –force* "'

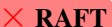

> The command to upgrade the IBM Storage Virtualize collection to the latest version from Ansible Galaxy is:
> "'*bash ansible-galaxy collection install -r requirements.yml* "'

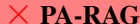

> "The command to upgrade the IBM Storage Virtualize collection to the latest version from Ansible Galaxy is:
> *ansible-galaxy collection install ibm.storage_virtualize –force*"

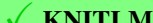

Only KNITLM provides the exact command that correctly upgrades the IBM Storage Virtualize collection. The instruct's response contains a typo in the collection name ('storwize' instead of 'storage_virtualize'), while RAFT and PA-RAG incorrectly rely on a requirements file, which is not specified in the ground truth.

## G    LLM USAGE

During the preparation of this manuscript, we employed a Large Language Model (LLM) as a writing support tool. Specifically, LLM was used to polish the phrasing, improve grammatical accuracy, and provide paraphrased alternatives to enhance clarity and readability. The LLM's role was limited to language refinement, and all suggested edits were reviewed and verified by the authors before inclusion.

