# OpenReview forum: "KnItLM: Weaving Knowledge into Instruction-Tuned LLMs via Continual Pre-Training and Merging"
_ICLR.cc/2026/Conference — Submitted to ICLR 2026_

### Official Review · Reviewer_Db8X · 2025-10-24

**Soundness:** 2
**Presentation:** 1
**Contribution:** 2
**Rating:** 2
**Confidence:** 4

**Summary:**

The KNITLM (Knowledge Ingestion via LoRA Merging) framework proposed in this paper addresses the core issues of "continuous pre training (CPT) disrupting instruction following ability" and "large dependence on synthetic data" in the domain knowledge injection of Instruction LLM. It innovatively adopts the technical route of "training knowledge LoRA on the Base LLM+integrating with Instruction LLM" to achieve collaborative retention of knowledge injection and instruction ability.

**Strengths:**

1. By training knowledge LoRA on the base model and integrating it with Instruction LLM, the synergy between "knowledge vector" and "instruction vector" is theoretically achieved using "task vector addition", which not only retains the instruction following ability of Instruction LLM, but also injects new domain knowledge.

2. By using the GRPO algorithm and a mixed reward of "effectiveness-efficiency-structural quality", the accuracy of knowledge injection is ensured while overfitting is suppressed, achieving lightweight training

**Weaknesses:**

1. In the relevant work section, although task arithmetic is mentioned, there is no in-depth comparison of the core differences between KNITLM and other model editing methods.

2. Using LLM as Jade binary scoring (0/1) as the core indicator, other key indicators of knowledge injection were not reported.

3. Verified only on two Redbook technical document datasets and not extended to other fields such as healthcare, finance, short text conversations, and long document reports. The model has not been tested on larger-scale models (20-80B) or different architecture models, making it impossible to determine the model compatibility of the method.

4. There are too few baselines used, and more SOTA models need to be compared.

**Questions:**

see above

---

> ### Author Response · Authors · 2025-11-18
>
> Dear reviewer Db8X
>
> Thank you for reviewing our work. It appears that there is some confusion in the understanding of our proposed approach, KnitLM. Strength#2 mentions “*By using GRPO algorithm and a mixed reward…*”; however, KnitLM has nothing to do with Reinforcement Learning (RL) or GRPO or rewards. We request that you kindly revisit this.
>
> Regarding the weaknesses, we make an attempt to address them -
>
> 1. **W1:** Please note that we are not proposing any new model-editing method (commonly known as model merging), and therefore, we do not delve into it in the related works section.  Rather, we are exploiting model merging for knowledge injection into an instruct model without compromising its instruction-following abilities.
> Direct CPT on the instruct model results in forgetting instruction following, and CPT on base requires redoing Instruction Fine Tuning (IFT). Our novelty lies in avoiding both by doing CPT on the base model and exploiting model-merging for imparting instruction following via instruct task vector obtained by subtracting weights of instruct and base model.
>
> 2. **W2: Why are we using LLM judge as the only indicator?** We are using QAs generated from the documents to determine if the knowledge has been successfully injected into the model. While there are other lexical and semantic similarity metrics such as RougeL-Recall, BertScore etc, as noted in [1],  none of them correlate with the human judgement. Only LLM Judge is shown to correlate with humans, and thus we focus only on the LLM judge metric. Nevertheless, we will soon report other metrics as well.
>
>    *[1] “Evaluating Correctness and Faithfulness of Instruction-Following Models for Question Answering”, TACL 2024*
>
>
>
> 3. **W3:** We are in the process of evaluating KnitLM on another dataset. Please note that we reported performance on two datasets and two models (Llama Instruct 8B and Mistral Instruct 7B).
>
>
> 4. **W4:** We have compared against two strong baselines for knowledge injection - RAFT and PA-RAG. Both require massive synthetic QA generation, whereas KnitLM can outperform them even with a fraction of QAs as it relies on the raw corpus for knowledge. Can you please recommend some baselines to compare against, which would further strengthen our work?

---

> > ### Author Response · Authors · 2025-11-21
> >
> > Dear Reviewer Db8X,
> >
> > As promised, we report the additional metrics and results to address your concerns.
> >
> > ## W2: Other metrics for knowledge ingestion
> >
> > We report word-level Recall (lexical) and BERTScore (semantic) for different knowledge ingestion methods.
> > Although KnitLM has the best recall in both QA and RAG setups, we wouldn’t rely on it, as we explained earlier in our response. As reported in [1], these metrics don’t correlate well with human judgment. On the other hand, LLM as a Judge is the only metric that correlates well with humans.  In a small human evaluation (reported in Appendix A.4–A.5), we also observe around 97% alignment between human and LLM judgements.
> >
> > Nevertheless, we will report Recall and BERTScore metrics for all experiments in the Appendix for transparency and reproducibility.
> >
> >
> > **Recall RedBook1 Mistral**
> >
> > |          |   QA  |       |      RAG     |              |
> > |----------|:-----:|:-----:|:------------:|:------------:|
> > |          |       |  All  | Ret. Success | Ret. Failure |
> > | Instruct | 58.49 | 71.22 |        77.81 |        62.54 |
> > | RAFT     | 56.81 | 71.46 |        77.87 |        63.00 |
> > | PA-RAG   | 58.72 | 71.76 |        77.47 |        65.55 |
> > | KnitLM   | 69.16 | 75.24 |        78.54 |        70.87 |
> >
> >
> >
> > **BertScore RedBook1 Mistral**
> >
> > |          |   QA  |       |      RAG     |              |
> > |----------|:-----:|:-----:|:------------:|:------------:|
> > |          |       |  All  | Ret. Success | Ret. Failure |
> > | Instruct | 35.68 | 50.11 |        57.03 |        40.99 |
> > | RAFT     | 41.21 | 57.18 |        62.66 |        49.95 |
> > | PA-RAG   | 47.08 | 57.40 |        59.47 |        54.66 |
> > | KnitLM   | 45.35 | 53.41 |        56.43 |        49.44 |
> >
> >
> >
> > **Recall RedBook2 Mistral**
> >
> > |          |   QA  |       |      RAG     |              |
> > |----------|:-----:|:-----:|:------------:|:------------:|
> > |          |       |  All  | Ret. Success | Ret. Failure |
> > | Instruct | 61.85 | 70.45 |        75.75 |        62.50 |
> > | RAFT     | 60.20 | 70.19 |        75.62 |        62.03 |
> > | PA-RAG   | 60.06 | 70.18 |        75.67 |        61.93 |
> > | KnitLM   | 64.88 | 71.77 |        76.32 |        64.92 |
> >
> >
> >
> >
> > **BertScore RedBook2 Mistral**
> >
> > |          |   QA  |       |      RAG     |              |
> > |----------|:-----:|:-----:|:------------:|:------------:|
> > |          |       |  All  | Ret. Success | Ret. Failure |
> > | Instruct | 27.44 | 46.48 |        53.72 |        35.61 |
> > | RAFT     | 37.90 | 53.19 |        60.08 |        42.85 |
> > | PA-RAG   | 37.86 | 52.98 |        59.90 |        42.58 |
> > | KnitLM   | 37.08 | 48.86 |        54.79 |        39.95 |
> >
> >
> >
> > ## W3: Evaluation on a different dataset
> >
> >
> > To address your concern regarding limited datasets in our experiments, we evaluate on an additional dataset, the QuALITY dataset (Pang et al., 2022) [2], which is different from the technical redbooks. It consists of many fictitious stories and test QAs from each story. We ingest the knowledge from a subset of 10 stories using KnitLM and evaluate the performance on test QAs. For training, we synthetically generate QAs from the stories while avoiding overlap with the test data. The table below compares KnitLM with RAFT.
> >
> > The main findings are as follows:
> > The out-of-the-box instruct model performs poorly on this dataset, only achieving 17.62% in the QA setup, confirming that the stories used for ingestion were never seen by the base model and that the dataset is a valid testbed for knowledge ingestion.
> > KnitLM obtains substantially higher QA and RAG performance than RAFT and the instruct baseline on this dataset. We see a 26% (43.22 to 54.61) increase from RAFT performance and an 8% (50.41 to 54.61) increase from the instruct model. We suspect the poor performance of RAFT could be due to its sensitivity to the size of training QAs. To further corroborate it, we will generate more training QAs and plot the performance of various methods against the training data size as already done in section 5.2 for redbooks.
> >
> > Dataset Statistics: \
> > Total words in the corpus: 43,254 \
> > Total training QA pairs generated: 2,460
> >
> >  We have omitted Pa-RAG due to time constraints. Recall that it requires substantially more synthetic data -- for each question, we need to generate multiple answers. Nevertheless, we are actively working on it and will provide updated results as soon as possible.
> >
> >
> > |          |   QA  |       |      RAG     |              |
> > |----------|:-----:|:-----:|:------------:|:------------:|
> > |          |       |  All  | Ret. Success | Ret. Failure |
> > | Instruct | 17.62 | 50.41 |        82.31 |        14.66 |
> > | RAFT     | 11.65 | 43.22 |        67.95 |        15.52 |
> > | KnitLM   | 25.75 | 54.61 |        83.85 |        21.84 |

---

> > > ### Author Response · Authors · 2025-11-21
> > > **Continuation to previous comment**
> > >
> > > ## W3: Evaluation on larger model sizes
> > >
> > > We report results on Qwen 2.5 32B using our proposed recipe.
> > >  The table below shows KnitLM consistently outperforms RAFT and the Instruct model across QA and RAG settings, confirming that KnitLM’s advantages generalise to different model families and larger parameter scales. We hope that it addresses your concern regarding generalization of our method across model sizes.
> > >
> > > |          |   QA  |       |      RAG     |              |
> > > |----------|:-----:|:-----:|:------------:|:------------:|
> > > |          |       |  All  | Ret. Success | Ret. Failure |
> > > | Instruct | 66.45 | 69.01 |        88.76 |        42.96 |
> > > | RAFT     | 68.37 | 82.75 |        91.01 |        71.85 |
> > > | KnitLM   | 74.12 | 86.26 |        95.51 |        74.07 |
> > >
> > >
> > >
> > >
> > > [1] “Evaluating Correctness and Faithfulness of Instruction-Following Models for Question Answering”, TACL 2024
> > >
> > > [2] Pang, Richard Yuanzhe, et al. "QuALITY: Question answering with long input texts, yes!." Proceedings of the 2022 Conference of the North American Chapter of the Association for Computational Linguistics: Human Language Technologies. 2022.

---

> > > > ### Comment · Reviewer_Db8X · 2025-11-27
> > > >
> > > > Thanks for your reply. I have updated my scores.

---

### Official Review · Reviewer_4Ptt · 2025-11-01

**Soundness:** 2
**Presentation:** 2
**Contribution:** 1
**Rating:** 2
**Confidence:** 3

**Summary:**

This paper addresses the challenge of incorporating domain-specific knowledge into instruction-tuned LLMs without compromising their instruction-following capabilities. The authors propose KnitLM, which performs continual pretraining (CPT) with LoRA adapters on base LLMs and transfers these adapters to instruction-tuned models. The authors also propose to use instruction model token embeddings during training to improve adapter compatibility.

**Strengths:**

- The paper tackles an important practical challenge in LLM: how to efficiently incorporate new domain knowledge without expensive instruction fine-tuning.
- The experimental setup carefully avoids data contamination, and the evaluations are reasonably thorough.

**Weaknesses:**

- The core contribution is incremental and somewhat limited. The central idea of combining LoRA-based continual pretraining with model merging (task arithmetic) is not novel. Prior work has explored task vector merging and LoRA transfer. The only substantive twist is using instruct-model token embeddings during LoRA training, which is a minor technical detail rather than a conceptual advance.
- The paper only compares against RAFT and PA-RAG. Stronger baselines (e.g., direct LoRA tuning on the instruct model, or parameter interpolation approaches) are missing.
- Section 3 uses $\Delta\theta$ to denote both task vectors (differences between model weights) and LoRA adapter weights. This is confusing because task vectors are full-rank weight differences, while LoRA adapters are low-rank decompositions.
- Minor comment: Citation formatting needs correction. Citations should appear in parentheses rather than as part of the text: Zhang et al. (2024b) -> (Zhang et al., 2024b). In the abstract, many abbreviations such as RAG,  LLM are used before defining them.

**Questions:**

1. Why not simply fine-tune base model fully on new knowledge (not LoRA), then compute true knowledge task vector and add to instruct model?

---

> ### Author Response · Authors · 2025-11-18
>
> Dear Reviewer 4Ptt,
>
> Thank you for reviewing our paper and for your constructive feedback. We address your concerns below -
>
> 1. **W1:** We would like to clarify that we are not claiming task vector merging or LoRA transfer as our contribution. Instead, we are applying task vector merging to achieve effective knowledge injection into an instruct model without compromising its instruction-following abilities.
> Direct CPT on the instruct model results in forgetting instruction-following, and CPT on base requires redoing Instruction Fine Tuning (IFT). Our novelty lies in avoiding both by doing CPT on the base model and exploiting model-merging for imparting instruction following via instruct task vector obtained by subtracting weights of instruct and base model.
> Competitive baselines for knowledge injection into an instruct model rely on generating massive synthetic QAs (as CPT on the instruct model using raw text doesn’t work). These QAs are the sole source of knowledge. On the other hand, KnitLM significantly reduces the need for synthetic data. It uses raw text as its source of knowledge and uses synthetic QAs to improve knowledge recall.
>
> 2. **W3 & Q1:** Use of $\Delta \theta$ for both task vector and LoRA adapter weights is deliberate. LoRA is just one way of obtaining the knowledge vector and task arithmetic is just one way of merging the task/knowledge vectors. We could replace LoRA with full fine-tuning of the base model, and also replace task-arithmetic with more sophisticated merging techniques such as TIES. The only thing that we lose is the modularity of the LoRA weights. If the knowledge vector is obtained via LoRA, and we use task-arithmetic for merging, we simply need to add it as an adapter over the pre-existing instruct model without actually merging the instruct vector and knowledge vector with the base model.
> We will soon report the performance with the above variants of KnitLM, where we use full finetuning for the knowledge vector and other methods like TIES [1] for model merging.
>
> 3. **W2:**  Both PA-RAG and RAFT are indeed direct LoRAs trained on the instruct model. The only difference is the training data. While PA-RAG and RAFT require passage(s)-question-answer triples to train, KnitLM uses raw text. Regarding parameter-interpolation approaches, do you mean merging techniques other than task-arithmetic, such as TIES [1]? As explained above, we will soon report performance with these variants of KnitLM.
>
>     *[1] Yadav, Prateek, et al. "Ties-merging: Resolving interference when merging models." Advances in Neural Information Processing Systems 36 (2023): 7093-7115.*
>
> 4. **W4:**  Thanks for pointing it out! We will fix it in the updated draft.

---

> > ### Author Response · Authors · 2025-11-21
> >
> > Dear Reviewer 4Ptt,
> >
> > As promised, we experimented with alternative model-merging methods (TIES [1]) and found similar performance gains as the simpler Task Arithmetic (TA) approach. Given TA’s simplicity, efficiency, and competitive performance, we recommend it as the default merging method. We include comparative results in the table below to support this claim.
> >
> > [1] Yadav, Prateek, et al. "Ties-merging: Resolving interference when merging models." Advances in Neural Information Processing Systems 36 (2023): 7093-7115.
> >
> > |              |   QA  |       |      RAG     |              |
> > |--------------|:-----:|:-----:|:------------:|:------------:|
> > |              |       |  All  | Ret. Success | Ret. Failure |
> > | Instruct     | 53.67 | 73.16 |        86.52 |        55.56 |
> > | KnitLM(TA)   | 73.80 | 86.58 |        92.13 |        79.26 |
> > | KnitLM(TIES) | 76.36 | 85.30 |        92.13 |        76.30 |

---

### Official Review · Reviewer_TYoE · 2025-11-01

**Soundness:** 2
**Presentation:** 3
**Contribution:** 3
**Rating:** 6
**Confidence:** 4

**Summary:**

This work presents a lightweight knowledge-ingestion method that performs CPT with LoRA on the base model and merges the knowledge adapter with the instruction model. An insightful technique is taking the instruction model's token embeddings during CPT to reduce distribution shift. The proposed method targets preserving the model's instruction-following ability while improving closed-book and RAG performance.

**Strengths:**

1. Training a knowledge LoRA on the base model and merging it with the instruction model is a lightweight technique that avoids re-running the SFT and may be RL for preserving the instruction following ability.
2. Using token embeddings from the instruct LLM during CPT is insightful and reasonable.

**Weaknesses:**

I am satisfied with the method itself, but have concerns about the experiment settings.
1. Limited baselines and benchmarks. The evaluation is restricted to two technical Redbooks. While the authors discuss test set quality and knowledge cutoff issues, the current scale is small and may not fully establish generality. The cutoff may be avoidable by testing the direct QA performance. If the model performs poorly for the test questions, it could imply that the model's parameterized knowledge does not contain the test set knowledge. In this way, this work can be comparable with board baseline models on more benchmarks.

**Questions:**

1. For the ablation study in section 5.4, the performance drop of e-KNITLM seems not very significant. The usefulness of replacing the embedding with the instruction model may contribute to two aspects, 1) as the author stated, avoiding OOD tokens like the special tokens used in the chat template, 2) for the tokens that are well-trained in the base model, the instruction model has a better representation. I am curious which one contribute more to the performance gain?
2. For the section 5.2, IMPACT OF THE SIZE OF THE SYNTHETIC DATA, I agree with the statement "access to QA from only a part of the corpus, will still show gains over the remaining data", but this seems to be a generic property. Does the KnItLM benifit more from this and why the baseline methods can not, since they also adopt synthetic data for training.

---

> ### Author Response · Authors · 2025-11-18
>
> Dear reviewer TYoE,
>
> We thank you for a constructive review of our manuscript and for raising a crucial concern. We will soon report performance on an additional dataset. Meanwhile, we answer your questions below -
>
> 1. **Q1: Usefulness of replacing the embedding with the Instruct model:** To clearly indicate the statistical significance of the gains, we will report confidence intervals in the updated manuscript. We would like to highlight that the gains from any knowledge injection method will be most visible in cases where the retriever fails to fetch the correct document. Under this scenario, we see a gain of 8.4% (73.13 to 79.26).
>
>    Thank you for suggesting an ablation to determine if the gains are due to (1) avoiding OOD tokens or  (2) better representation of the other well-trained tokens.  We will soon report our observations.
>
>
>
> 2.  **Q2**
> > For the section 5.2, IMPACT OF THE SIZE OF THE SYNTHETIC DATA, I agree with the statement "access to QA from only a part of the corpus, will still show gains over the remaining data", but this seems to be a generic property.
>
>
>     You are absolutely right that this is a generic property of LLMs, which KnitLM is able to exploit by virtue of using the entire corpus during CPT.  A major issue with SFT-based methods, such as RAFT and PA-RAG, is that they rely solely on synthetically generated QAs for knowledge injection. On the other hand, KnitLM learns from the raw documents as well. Imagine a scenario where we generate synthetic QAs for training only from the first half of the document. RAFT and PA-RAG would have no mechanism to ingest knowledge from the second half, whereas KnitLM would be able to do so via CPT on the entire document. In short, KnitLM doesn’t rely on synthetic QAs for knowledge injection, whereas synthetic QAs are the only source of knowledge for the SFT-based baselines. Hence, KnitLM is more robust to the amount of synthetic training QAs.
>
>
>
> Please let us know if you have any other concerns, and we will be happy to address them.

---

> > ### Author Response · Authors · 2025-11-21
> >
> > Dear Reviewer TYoE,
> >
> > To address your concern regarding limited datasets in our experiments, we evaluate an additional dataset, the QuALITY dataset (Pang et al., 2022) [2], which is different from the technical redbooks. It consists of many fictitious stories and test QAs from each story. We ingest the knowledge from a subset of 10 stories using KnitLM and evaluate the performance on test QAs. For training, we synthetically generate QAs from the stories while avoiding overlap with the test data. The table below compares KnitLM with RAFT.
> >
> > The main findings are as follows:
> > - The out-of-the-box instruct model performs poorly on this dataset, only achieving 17.62% in the QA setup, confirming that the stories used for ingestion were never seen by the base model and that the dataset is a valid testbed for knowledge ingestion.
> > - KnitLM obtains substantially higher QA and RAG performance than RAFT and the instruct baseline on this dataset. We see a 26% (43.22 to 54.61) increase from RAFT performance and an 8% (50.41 to 54.61) increase from the instruct model. We suspect the poor performance of RAFT could be due to its sensitivity to the size of training QAs. To further corroborate it, we will generate more training QAs and plot the performance of various methods against the training data size as already done in section 5.2 for redbooks.
> >
> > Dataset Statistics: \
> > Total words in the corpus: 43,254 \
> > Total training QA pairs generated: 2,460
> >
> >  We have omitted Pa-RAG due to time constraints. Recall that it requires substantially more synthetic data -- for each question, we need to generate multiple answers. Nevertheless, we are actively working on it and will provide updated results as soon as possible.
> >
> >
> > |          |   QA  |       |      RAG     |              |
> > |----------|:-----:|:-----:|:------------:|:------------:|
> > |          |       |  All  | Ret. Success | Ret. Failure |
> > | Instruct | 17.62 | 50.41 |        82.31 |        14.66 |
> > | RAFT     | 11.65 | 43.22 |        67.95 |        15.52 |
> > | KnitLM   | 25.75 | 54.61 |        83.85 |        21.84 |
> >
> >
> >
> >
> > |          |   QA  |       |      RAG     |              |
> > |----------|:-----:|:-----:|:------------:|:------------:|
> > |          |       |  All  | Ret. Success | Ret. Failure |
> > | Instruct | 17.62 | 50.41 |        82.31 |        14.66 |
> > | RAFT     | 11.65 | 43.22 |        67.95 |        15.52 |
> > | KnitLM   | 25.75 | 54.61 |        83.85 |        21.84 |

---

> ### Author Response · Authors · 2025-11-29
>
> Dear Reviewer TYoE,
>
> As promised, we decided to dig deep into the efficacy of embedding swap during training. Specifically, we conducted experiments to determine where the major performance boost originates during embedding swap: from the OOD tokens in the base model, such as those introduced in the chat template, or from the tokens already trained in the base model. We use two strategies to automatically select the probable OOD tokens:
> - top-k: Select top-k tokens w.r.t. the L-2 norm of the difference between instruct and base models’ token embeddings
> - top-p: A nucleus sampling-like approach where we select the top-p tokens upto a threshold of 0.9.
>
> To confirm that they indeed contain the OOD tokens, we paste them below the table for both Llama and Mistral.
> Row 2 and 3 (e-50-KniTLM & top-p-0.9) represents the version where only the most different embeddings were swapped, and Row 4 and 5 (ex-50-KniTLM & bottom-p-0.1) represents the version where the most different embeddings were excluded, and only the rest of the embeddings were swapped. Recall that e-KnitLM (Row  6) represents the version where none of the embeddings were swapped.
>
> |              |   QA  |       |      RAG      |               |
> |--------------|:-----:|:-----:|:-------------:|:-------------:|
> |              |       |  All  | Ret.  Success | Ret.  Failure |
> | KnitLM       | 73.80 | 86.58 |         92.13 |         79.26 |
> | e-50-KniTLM       | 74.12 | 87.22 |         94.38 |         77.78 |
> | top_p_0.9    | 74.12 | 84.66 |         91.01 |         76.30 |
> | ex-50-KniTLM | 72.52 | 83.39 |         92.13 |         71.85 |
> | bottom_p_0.1 | 73.48 | 81.79 |         88.20 |         73.33 |
> | e-KnitLM     | 72.76 | 84.57 |         92.13 |         73.13 |
>
> We see that clearly the top-50 embeddings influence the final trained model much more than the rest, achieving performance equivalent to the full KnitLM regime with just the top 50 embeddings swapped. On the other hand, we can see Row 4 (ex-50-KniTLM) and Row 6 (e-KnitLM) having comparable performance indicating that the embeddings which were the same in both the instruct and base models have little impact on the final performance. Another interesting observation is the drop in performance between rows 2(e-50-KnitLM) and 3(top-p-0.9). One would expect that adding more meaningful tokens from the instruct model would improve performance, this is not what we see. Our hypothesis as to why this happens can be explained due to 2 major phenomena:
>
> - the inherent value of each token from the instruct model i.e. how useful a particular token is to the training
> - the consistency of the entire embedding layer i.e., how consistent the embedding layer is with respect to its tokens (an embedding layer with only tokens from one model is said to be highly consistent where as a layer with 50% tokens from one model and the rest from another is said to be highly inconsistent).
>
> We posit that the OOD tokens help to a certain extent but soon start interfering with the other tokens as inconsistency within the layer increases. However, swapping the entire layer preserves consistency(as all tokens in the base model are swapped with instruct model’s) as well as utilising the more useful tokens during training. As swapping does not introduce any computational bottleneck we advise to always switch the entire layer as opposed to targeted tokens.
>
> Please see below for the list of OOD tokens for the base model:
>
> Most different tokens (Llama)
>
> 0. <|end_header_id|>
> 1. <|start_header_id|>
> 2. <|end_of_text|>
> 3. assistant
> 4. <|eot_id|>
> 5. <|eom_id|>
> 6. ============Ċ
> 7. ========Ċ
> 8. ===========Ċ
> 9. .scalablytyped
> 10. ?"ĊĊĊĊ
> 11. ĠâĢĿĊĊ
> 12. <|python_tag|>
> 13. Ġanalsex
> 14. =========Ċ
> 15. ==========Ċ
> 16. âĢ¦"ĊĊ
> 17. âĢ¦âĢ¦âĢ¦âĢ¦âĢ¦âĢ¦âĢ¦âĢ¦
> 18. ..ĊĊĊĊ
> 19. ===============Ċ
>
> Mean of norms of top 50 tokens: 0.151 (26.5x)
>
> Mean of norms of bottom 127950 tokens: 0.0057
>
> Most different tokens (Mistral)
>
> 0. 
> 1. ‑
> 2. :%.*
> 3. :%.*]]
> 4. eltemperaturen
> 5. tagHelper
> 6. ❒
> 7. ederbörd
> 8. ▁listade
> 9. ▁/******/
> 10.
> 11. ▁#!
> 12. ▁{
> 13. \<s\>
> 14. ⌁
> 15. ICENSE
> 16. ————
> 17. ▁SDValue
> 18. ERCHANTABILITY
> 19. ▁febbra
> 20. qpoint
>
> Mean of norms of top 50 tokens: 0.02 (4.76x)
>
> mean of norms of bottom 32718 tokens: 0.0042

---

### Official Review · Reviewer_saay · 2025-11-03

**Soundness:** 3
**Presentation:** 3
**Contribution:** 3
**Rating:** 6
**Confidence:** 4

**Summary:**

This paper proposes a novel continued pre-training approach for incorporating new knowledge into instruction-tuned LLMs, addressing the issue that conventional methods often degrade instruction-following capabilities. The authors introduce a method that merges two task vectors: a knowledge vector (representing the new knowledge to be injected) and an instruction task vector, obtained as the difference between the baseline model vector and the instruction-tuned model vector. Experimental results show that KnItLM achieves improvements over existing baseline methods.

**Strengths:**

1.	The proposed merging of the instruction-following vector and the knowledge vector based on task vectors is novel, well motivated, and technically interesting.
2.	The special treatment of token embeddings adds further depth and elaboration to the method.
3.	The paper is clearly written and easy to follow, with detailed explanations.
4.	The experimental results demonstrate that the proposed methods achieve performance improvements over the baselines.

**Weaknesses:**

1.	In Table 1, it is unclear whether the reported improvements are substantial.
2.	It is not clear whether the proposed method can be applied in an incremental manner when additional knowledge is introduced. Once the knowledge vector is injected into the base model, does the updated model then serve as the new base model for the next stage?
3.	Additional comparisons with other continual learning baselines would strengthen the experimental evaluation.

**Questions:**

Please see weaknesses.

---

> ### Author Response · Authors · 2025-11-18
>
> Dear reviewer saay,
>
> Thank you for your positive review of our manuscript. Below, we make a preliminary attempt to address your concerns and initiate a discussion.
>
> 1. **Are improvements substantial in Table 1:** We test the performance under two settings -- QA and RAG. In the former, LLM is provided with only the question, and no retrieved context is provided. In this setting, substantial performance gains are observed for both datasets. Specifically, we observe gains of 15% (64.22 to 73.88) and 50% (27.23 to 40.98), respectively, for the two datasets over the baselines.
>
>    In the RAG setup, LLM is provided with the top 5 retrieved passages. Since Instruct LLMs are generally good at reading comprehension tasks, we expect them to perform well when the retriever is able to fetch the passage(s) containing the gold answer. We expect knowledge injection methods to give substantial gains when the retriever fails to fetch gold passages. In this scenario, we observe gains of 7% (74.07 to 79.26) and 22.5% (37.64 to 46.13), respectively, for the two datasets compared to the baselines.
>
>    All the reported gains are statistically significant. To make this evident, we will add confidence intervals around the average reported performance in the updated manuscript.
>
> 2. **Application in an incremental manner:** Thank you for asking this question! We did not consider our approach as a continual learning method. Instead, we proposed KnitLM as a single-step process for injecting new knowledge while preserving instruction following. Inspired by your comment, we are running experiments to compare different strategies for incremental knowledge injection. We will soon report the results in a separate comment.
>
> 3.  **Comparison with other continual learning baselines:** As explained above, our approach is not a continual learning method, and therefore, we do not compare it against those methods. We understand that the confusion might have stemmed from the use of  “Continual pretraining” in the title. We will replace it with “continued pretraining” to avoid confusion. Nevertheless, if you have a particular baseline in mind, please let us know, and we will be happy to evaluate and compare it against our method.

---

> ### Author Response · Authors · 2025-11-21
>
> Dear Reviewer saay,
> Your comment on the potential use of KnitLM for continual learning piqued our interest and we decided to run a small experiment trying to test whether our method is a viable option for continual learning. We will first describe the experimental setup then dive into the results and their plausible explanations.
>
> Experiment Setup:
> - Train a LoRA on a dataset and merge that LoRA into the base model. For example in the second row of the tables below, we first train the adapter on RedBook 1 (R1) and merge it into the base model.
> - Then train a new LoRA on the next dataset (so the base already contains the previously trained weights merged into it). For the second row of our tables,  we train a new lora on RedBook2 (R2) over the merged model mentioned in step 1.
> - Depending on the order we choose, we get two trained models -- R1→R2 and R2→R1.
> - In the table below, we compare these two models with the ones trained separately for the two datasets.
>
> Main Findings:
> - As seen in the table below, continual training in this naive fashion leads to significant drop in performance as compared to training separate models for the datasets.
> - Comparing R1→R2 and R2→R1,  the former performs better on 2nd dataset and the later performs better on 1st dataset, implying that the model forgets the knowledge ingested earlier. Thus, as we suspected, this method is not conducive for continual learning and studying that is beyond the scope of the current work.
>
>
>
> |        |       |       |   RedBook 1  |              |       |       |   RedBook 2  |              |
> |--------|:-----:|:-----:|:------------:|:------------:|:-----:|:-----:|:------------:|:------------:|
> |        |   QA  |       |      RAG     |              |   QA  |       |      RAG     |              |
> |        |       |  All  | Ret. Success | Ret. Failure |       |  All  | Ret. Success | Ret. Failure |
> | KnitLM | 73.80 | 86.58 |        92.13 |        79.26 | 40.98 | 66.86 |        80.67 |        46.13 |
> | R1->R2 | 48.56 | 73.16 |        88.76 |        52.59 | 36.36 | 64.61 |        80.71 |        40.42 |
> | R2->R1 | 68.69 | 81.79 |        93.26 |        66.67 | 20.66 | 63.51 |        79.85 |        38.97 |

---

### Author Response · Authors · 2025-11-29
**Common Response to All Reviewers**

Dear Reviewers,

Thank you for the thoughtful and constructive feedback. We appreciate Reviewers saay, TYoE, and Db8X for highlighting the novelty and motivation of merging knowledge and instruction vectors through lightweight LoRA-based continued pretraining. We also thank Reviewers saay and TYoE for noting the insight and practical value of using instruction-model token embeddings to reduce distribution shift. Below we provide a concise, collated summary of the major new experiments.

# New model size and family (Qwen 2.5 32B)

We applied the KnitLM recipe to a larger model family/scale (Qwen 2.5 32B) to verify that the method generalizes across parameter scale and model family. The results are given in Table 1:

Table 1: Performance on Book 1 test set using Qwen 2.5 32B model.
|          |   QA  |       |      RAG     |              |
| -------- | :---: | :---: | :----------: | :----------: |
|          |       |  All  | Ret. Success | Ret. Failure |
| Instruct | 66.45 | 69.01 |     88.76    |     42.96    |
| RAFT     | 68.37 | 82.75 |     91.01    |     71.85    |
| KnitLM   | 74.12 | 86.26 |     95.51    |     74.07    |

# New non-technical dataset (QuALITY)

We evaluated KnitLM on QuALITY (10 stories; 43,254 words; 2,460 synthetic QA pairs) to test domain robustness beyond technical redbooks. Results can be found in Table 2:

Table 2: Comparative performance of KnitLM on the QuALITY dataset.
|          |   QA  |       |      RAG     |              |
| -------- | :---: | :---: | :----------: | :----------: |
|          |       |  All  | Ret. Success | Ret. Failure |
| Instruct | 17.62 | 50.41 |     82.31    |     14.66    |
| RAFT     | 11.65 | 43.22 |     67.95    |     15.52    |
| KnitLM   | 25.75 | 54.61 |     83.85    |     21.84    |

# Impact of token swaps during training

We conducted experiments to determine where the major performance boost originates during embedding swap: from the OOD tokens in the base model, such as those introduced in the chat template, or from the tokens already trained in the base model. We employ two strategies to automatically select the probable OOD tokens, selecting the top-k tokens w.r.t. the L-2 norm of the difference between instruct and base models’ token embeddings and a nucleus sampling-like approach where we select the top-p tokens upto a threshold of 0.9. For more details about the experiment and a run down of the results please refer to the comment posted for reviewer TYoE
Results are provided in Table 3:

Table 3: Impact of partial token swapping during training.

|              |   QA  |       |      RAG      |               |
| ------------ | :---: | :---: | :-----------: | :-----------: |
|              |       |  All  | Ret.  Success | Ret.  Failure |
| KnitLM       | 73.80 | 86.58 |     92.13     |     79.26     |
| e-50-KniTLM  | 74.12 | 87.22 |     94.38     |     77.78     |
| top_p_0.9    | 74.12 | 84.66 |     91.01     |     76.30     |
| ex-50-KniTLM | 72.52 | 83.39 |     92.13     |     71.85     |
| bottom_p_0.1 | 73.48 | 81.79 |     88.20     |     73.33     |
| e-KnitLM     | 72.76 | 84.57 |     92.13     |     73.13     |

# Effect of other model-merging strategies (Task Arithmetic vs TIES)

We compared our default Task Arithmetic (TA) merging with TIES to see if more sophisticated merging yields consistent gains. Results can be found below:

Table 4: Impact of different merging techniques
|              |   QA  |       |      RAG     |              |
| ------------ | :---: | :---: | :----------: | :----------: |
|              |       |  All  | Ret. Success | Ret. Failure |
| Instruct     | 53.67 | 73.16 |     86.52    |     55.56    |
| KnitLM(TA)   | 73.80 | 86.58 |     92.13    |     79.26    |
| KnitLM(TIES) | 76.36 | 85.30 |     92.13    |     76.30    |

We will include these consolidated tables and brief interpretations in the revised manuscript, add confidence intervals for reported averages, and update wording (e.g., “continued pretraining”) to avoid confusion with continual learning. Thank you again, your feedback directly shaped these experiments and clarified the scope and strengths of KnitLM.

Warm regards,

The authors

---

### Meta-Review · Area_Chair_J4BP · 2026-01-09

**Summary:**

This manuscript was evaluated by four experts in the field, resulting in recommendation scores of (2, 2, 6, 6). The consensus among reviewers is that the paper is not currently ready for publication and requires significant revision, particularly regarding experimental evaluation, presentation, and comparative analysis. Reviewer saay noted that while the reported improvements appear substantial, additional experimental comparisons are missing. Reviewer TYoE expressed serious concerns regarding the limited baselines and benchmarks, noting that the performance improvements are not statistically significant. Reviewer 4Ptt described the core contribution as incremental and somewhat limited, highlighting missing comparisons and the need for clearer presentation. Reviewer Db8X pointed out that numerous related works were not discussed and several necessary experiments were omitted. The manuscript requires further polishing to make the motivation clearer and more convincing. Based on these concerns, the paper cannot be accepted at this time. The authors are encouraged to address the reviewers' comments when revising the paper for future submission.

**Reviewer Concerns:**

Although the rebuttal effectively resolved issues regarding presentation and missing references, the experimental evaluation remains a critical shortcoming. The additional comparisons provided in the revision are inadequate, and the current empirical evidence does not convincingly demonstrate the proposed method's advantage over established baselines.

**Reviewer Scores:**

The reviewers might acknowledge the authors' commitment to incorporating missing references and refining the presentation in the revised version. However, significant concerns persist regarding the experimental evaluation and the depth of analysis.

---

### Decision · Program_Chairs · 2026-01-26

Reject